# Effects of Maltodextrin and Gum Arabic Composition on the Physical and Antioxidant Activities of Dewaxed Stingless Bee Cerumen

**DOI:** 10.3390/foods12203740

**Published:** 2023-10-11

**Authors:** Nuha Binte Iesa, Supakit Chaipoot, Rewat Phongphisutthinant, Pairote Wiriyacharee, Bee Gim Lim, Korawan Sringarm, Michael Burgett, Bajaree Chuttong

**Affiliations:** 1Chemical Engineering and Food Technology Cluster, Singapore Institute of Technology, 10 Dover Drive, Singapore 138683, Singapore; 1801185@sit.singaporetech.edu.sg (N.B.I.); beegim.lim@singaporetech.edu.sg (B.G.L.); 2Meliponini and Apini Research Laboratory, Department of Entomology and Plant Pathology, Faculty of Agriculture, Chiang Mai University, Chiang Mai 50200, Thailand; michael.burgett@oregonstate.edu; 3Multidisciplinary Research Institute, Chiang Mai University, Chiang Mai 50200, Thailand; supakit.ch@cmu.ac.th (S.C.); rewat.p@cmu.ac.th (R.P.); 4Division of Product Development Technology, Faculty of Agro-Industry, Chiang Mai University, Chiang Mai 50100, Thailand; pairote.w@cmu.ac.th; 5Department of Animal and Aquatic Sciences, Faculty of Agriculture, Chiang Mai University, Chiang Mai 50200, Thailand; korawan.s@cmu.ac.th; 6Department of Horticulture, Oregon State University, Corvallis, OR 97331, USA

**Keywords:** antioxidant activity, carrier ratio, cerumen, encapsulation, stingless bees

## Abstract

Background: Cerumen is a mixture of beeswax and plant resin made by stingless bees. It has antimicrobial and antioxidant properties and is often used in biological and therapeutic treatments. However, its adhesive characteristic makes cerumen challenging to process into powder. Methods: This study investigated the physical characteristics and antioxidant activity of the encapsulated freeze-dried dewaxed cerumen of *Tetragonula laevicpes*. The combination of coating materials at concentrations of 20%, 30% and 40% and carrier ratios of maltodextrin to gum arabic of 9:1, 5:5 and 3:7 were used to encapsulate dewaxed cerumen when freeze-dried; the control was maltodextrin at a concentration of 31.25%. Results: All carrier matrices showed high yields of >80% and similar powder characteristics of low moisture content, low water activity, high glass transition temperature and water dispersibility. Overall, antioxidant activities ranged from 69–80%, while the encapsulation efficiency of total phenolic content ranged from 46–68%. All carrier matrices show higher antioxidant activities than 31.25% maltodextrin, with the lowest antioxidant at 57%. Conclusions: The carrier ratio of 5:5 resulted in better physical properties and retained 68% of polyphenolic activity in powders.

## 1. Introduction

Stingless bees are the largest group of eusocial bees, with 605 described species [1]. They can be found in the tropical and subtropical regions of the world [2], as they are eusocial insects that collect food from plants and store it in a hive comparable to that of the honey bee (*Apis mellifera*). The stingless bees store their food (honey and pollen) in a waxy-resinous material called storage pots made from cerumen, not in hexagonal comb cells like the honey bees. Cerumen consists of bee secretions (wax and saliva) and resinous plant material produced by stingless bees. Some meliponines occasionally mix the cerumen with clay, soil or other substances, making its texture less resinous than propolis [3]. In this article, the stingless bee species *Tetragonula laeviceps* is referred to as a cryptic species group, the *T. laeviceps* species complex. *T. laeviceps* is generally distributed across Southeast Asia [4]. Due to their opportunistic behavior, *T. laeviceps* is the most commonly manipulated species in wooden box hives by beekeepers for honey and cerumen production [4]. The constituents of cerumen differ depending on bee species, available plant sources, geographical locations and seasonal changes. Honey bees produce propolis, whereas stingless bees produce propolis, geopropolis and cerumen [5]. Traditional medicine has integrated stingless bee products like honey, pollen, propolis, geopropolis and cerumen. Its historical utilization, spanning various cultures from ancient societies to the present day, emphasizes its importance. Stingless bee propolis, abundant in compounds and possessing strong antimicrobial attributes, has found its place in diverse folk remedies, including those of Western Maharashtra, India [6,7]. Throughout history, propolis has served diverse purposes across cultures. The Greeks, Romans, Egyptians, Arabians, and Incas all valued its healing properties. Recognized as an official drug in 17th-century London, its antibacterial traits fueled its popularity. It aided wound treatment in World War II and gained approval for varied uses, even addressing tuberculosis in 1969 [8,9]. Cerumen is known for its biological and therapeutic properties and is generally used in traditional remedies to treat wounds, colds, ulcers, skin disorders, and general health stimulants. It has been considered an alternative supplementary treatment for virus infection, is extensively accessible, and rarely causes side effects [10,11,12]. Stingless bee products have religious value in some indigenous cultures and are used in traditions and ceremonies. Honey, cerumen, and stingless bee nests may be utilized as presents or symbolic objects in religious activities [11,13]. Cerumen from stingless bees, along with materials like plant fibers, can find application in traditional crafts and construction [14,15].

The solvent utilized in propolis extraction plays a vital role in determining the quality of propolis obtained from raw propolis, wax, and other impurities. This is because the dissolution of diverse compounds in various solvents is contingent upon their polarity. Numerous solvents, such as ethanol, methanol, propylene glycol, water and edible oils, have been investigated for the extraction of propolis. Specifically, ethanolic extraction is most often used because of its ability to achieve low-wax propolis extracts rich in bioactive components [16,17]. Microencapsulation of propolis with the use of powder technology such as spray-drying and freeze-drying has been undertaken to broaden the application of propolis, counter its residual solid flavor and aroma, and obtain an alcohol-free propolis ingredient [18]. Both techniques have proven their potential to produce propolis powders with high levels of encapsulated phenols, low water activities, good solubility in cold water and high antioxidant activities. Encapsulation materials or carriers play a vital role in the encapsulation of propolis bioactive components to minimize their undesirable flavor and aroma [19]. Thus far, very little information on the characteristics of dewaxed cerumen powder has been published after extraction, and the optimal effect of carriers has not been explored. Therefore, the objectives of this study were to examine the freeze-drying process of dewaxed cerumen coating materials, evaluate their powder characteristics and examine the variations in maltodextrin-gum arabic concentrations and ratios to determine the optimal matrix with the best physical and antioxidant activity of the encapsulated dewaxed cerumen. In response to the increase in the benefits of *Tetragonula* spp. products, research on the species is still scarce compared to *A. mellifera* propolis.

## 2. Materials and Methods

### 2.1. Raw Material and Chemicals

The cerumen of *Tetragonula laeviceps* was purchased from the meliponiculturists in Ramphan village, Tha Mai District, Chanthaburi 22170, Thailand, and shipped to the Singapore Institute of Technology, Singapore. Gum Arabic Grade 1 (Hashab Acacia Senegal) was acquired from Ingredion Pte Ltd., Singapore, and Maltodextrin DE 10–20 was purchased from Suntop Enterprise Pte Ltd., Singapore. Folin–Ciocalteu’s phenol reagent, 1,1-Diphenyl-2-picrylhydrazyl Free Radical (DPPH), gallic acid monohydrate ACS reagent, sodium carbonate anhydrous, and analytical grade ethanol (99.9%) were obtained from Sigma-Aldrich, Singapore.

### 2.2. Cerumen Dewaxing

#### 2.2.1. Separation of Wax from Cerumen

Modified from Krell (1996) [20], cerumen was first prepared by removing debris and then kept in −18 °C freezer overnight. Frozen cerumen was chopped into 2–4 cm sections and washed using 25 °C deionized (DI) water for 1 min and repeated washing 2 times in a 70-micron sieve. Washed cerumen was transferred onto aluminum trays under constant airflow until fully dried. After drying, the cerumen was placed into plastic containers and stored in a −18 °C freezer overnight. The frozen chopped cerumen was mixed with liquid nitrogen (approximately 200 g with 20 mL of liquid nitrogen) and blended in a stainless-steel Waring Blender at high speed for 30 s in intervals of 10 s to prevent large clumps of cerumen. The powdered cerumen was then transferred into containers and stored in a −18 °C freezer. The separation of wax from cerumen was done by heating and centrifugation. 250 g of powdered cerumen was heated with DI water (1 g:2.5 mL) on a hot plate at 90 °C for 1 min to melt the available wax. The heated mixtures were inserted into a water bath (Memmert) at 60 °C for 3 h until a distinct top layer of wax and a bottom layer of dewaxed cerumen (bottom layer) could be seen. The wax layer and a large amount of water were first removed, and the sedimented layer was transferred into 500-mL centrifuge tubes. The tubes were centrifuged at 2985 RCF/3011 rpm using Sigma 8KS, (Sigma Laborzentrifugen GmbH, Osterode am Harz, Germany) for 15 min. The supernatant was disposed of, and the precipitate was collected as crude dewaxed cerumen, which was transferred into containers and stored in a −18 °C freezer.

#### 2.2.2. Ethanolic Extraction of Dewaxed Cerumen

Adapted from Kubiliene et al. (2015) [17], dewaxed cerumen was mixed with 70% technical-grade ethanol (1 g:10 mL) into volumetric flasks and covered with parafilm. The flasks were macerated at 25 °C and 250 rpm for 48 h (Thermo Scientific™ MaxQ™ 6000, Waltham, MA, USA). After 48 h, the solution was filtered using Whatman filter paper no. 1 under vacuum to obtain a clear orange filtrate. The collected filtrate was stored in dark glass bottles and kept at −4 °C in the refrigerator until further processing.

#### 2.2.3. Removal of the Ethanolic Solvent from Dewaxed Cerumen

Three drying methods have been applied to maximize the removal of ethanol: rotary evaporator, air drying and vacuum oven drying. The mass of ethanol was first evaporated using a rotary evaporator (Buchi Rotary Evaporator R-300, Flawil, Switzerland) at a vacuum pressure of 40 mbar, a heating bath temperature at 50 °C, coolant temperature at 10 °C and 25 rpm rotation speed. The concentrated dewaxed cerumen was then divided into 150 mL glass beakers. The beakers were set in a fume hood (Kewaunee Scientific, Statesville, NC, USA) at a flow rate of 0.96 m/s for 2 h and transferred into a vacuum oven (Binder, Tuttlingen, Germany) overnight at 38 °C for 15 h, followed by 50 °C for 2 h, until all ethanol has been removed.

### 2.3. Encapsulation of Dewaxed Cerumen by Freeze-Drying

In Šturm et al. (2019) [18], The propolis to carrier ratio weight of 1:4 (*w*/*w*) and the formulations of maltodextrin-gum arabic carriers were prepared at mass ratios (*w*/*w*) of the solid content of 9:1, 5:5 and 3:7. The concentration of carriers for samples was prepared at 20%, 30% and 40% with a control of maltodextrin at a carrier concentration of 31.25%, as referenced in [18]. The required water weight was first added into beakers with the dewaxed cerumen of the respective samples, then double-boiled at 40 °C for 15 min to soften the dewaxed cerumen paste. The solution was mixed with the required carriers (Silverson L5M-A, Selangor, Malaysia) at 4300 rpm for 5 min in a double boiler. The solutions were heated to 60 °C and held for 1 min. The homogenous samples were poured into 50-mL polyethylene terephthalate (PET) centrifuge tubes and pre-frozen at −80 °C overnight. The centrifuge caps were replaced with a light-duty wipe (Kimtech Kimwipe, Bryan, TX, USA) and freeze-dried (Virtis Benchtop Pro, York, UK) in auto mode at −80 °C with a fixed condenser threshold setpoint of −40 °C for 3–4 days. The dried samples were placed into dark glass bottles and pounded using a pestle for 3 min until fine powders were obtained.

### 2.4. Glass Transition POINT Measurement

The glass transition temperature (Tg) was determined by a differential scanning calorimeter (NETZSCH DSC 214 Polyma, Selb, Germany). Powdered samples of ca. 10–15 mg were conditioned in aluminum pans with inverted caps, hermetically sealed and heated at 10 °C/min from −40 to 90 °C, in an inert temperature (45 mL/min of N_2_). An empty pan was used as a reference. The condition of concentrated ethanolic dewaxed cerumen pastes before the freeze-drying was similarly conditioned into the aluminum pans with a hole on the inverted cap. This hole would allow excess moisture from the dewaxed cerumen paste to be removed while heating. An empty pan with a holed lid was used as a reference. The mid-point values for Tg of the samples were taken. Triplicates were carried out.

### 2.5. Particle Size Measurement

#### 2.5.1. Scanning Electron Microscope

The morphology of freeze-dried particles was observed using a scanning electron microscope (JEOL JSM-IT300, Tokyo, Japan) at an accelerating voltage of 5–10 kV. The powdered samples were mounted on sample holders using double-sided carbon tape and sputtered with platinum for 60 s at 20 mA using a fine auto coater (JEOL JEC-3000FC, Tokyo, Japan). The platinum-coated samples were subsequently viewed under the microscope at a magnification of 400×. Duplicates were carried out.

#### 2.5.2. Light Microscopy of Dewaxed Cerumen Solution

The stock solution of dewaxed cerumen powder in DI water was prepared at 20 mg/mL. The solutions were vortexed at 1800 rpm before analysis. The solutions were analyzed under a light microscope (NIS-Elements Ar 4.60.00 64-bit, Nikon Eclipse Ci, Melville, NY, USA) at magnifications of 10× and 100× after being dissolved in solutions. Duplicates were carried out.

#### 2.5.3. Particle Size Measurements

The stock solution of dewaxed cerumen powders in DI water was prepared at 10 mg/mL. The solutions were vortexed at 1800 rpm before analysis. The particle size distribution and the particle diameter, D (4,3) (volume or mass moment mean), were measured using Mastersizer 3000 (Hydro MV, Malvern Panalytical, Westborough, MA, USA) at a refractive index of 1.36, stirrer speed at 3000 rpm, and 80% ultrasound to break up particles. Measurements were carried out in duplicate at 25 °C.

### 2.6. Physiochemical Characterization of Powders

#### 2.6.1. Water Dispersibility (WD)

The water dispersibility method was adapted from Šturm et al. (2019) [18] by suspending 0.5 g of dewaxed cerumen powder in 50 mL of distilled water and vortexed (Vortex Gene2, Scientific Industries, Bohemia, NY, USA) for 5 min at 8000 rpm. The solutions were centrifuged (Eppendorf Centrifuge 5430R, Hamburg, Germany) at 3075 RCF for 10 min, then 20 mL supernatant aliquots were dried in a Memmert air oven at 105 °C for 3 h and 40 min. All measurements were carried out in duplicate. The water dispersibility was calculated according to Equation (1).
(1)$$\mathrm{WD}\;\left[\%\right]=\frac{\mathrm{weight}\;\mathrm{of}\;\mathrm{solid}\;\mathrm{supernatent}\;\left[g\right]\times 2.5}{\mathrm{weight}\;\mathrm{of}\;\mathrm{sample}\;\left[g\right]}\times 100$$

#### 2.6.2. Water Activity (Aw)

Water activity was analyzed using a water activity meter (Aqualab TDL Benchtop Water Activity Meter, Haryana, India) with an accuracy of ±0.003. The equipment was calibrated with standard LiCl salt solutions of 0.250 Aw, 0.500 Aw, and 0.760 Aw before determining the freeze-dried dewaxed cerumen powders’ water activities. Measurements were done in duplicate with approximately 1.5 g of freeze-dried powders at 25 °C.

#### 2.6.3. Moisture Content (MC)

A Mettler Toledo infrared moisture analyzer was used to determine the moisture content of approximately 2 g of freeze-dried powders at 105 °C. Measurements were done in duplicate.

### 2.7. Antioxidant Activity Assays

#### 2.7.1. Free Radical-Scavenging Activity Measurement

A concentration of 0.1 mM DPPH in a 96% ethanolic solution was prepared, and 2.9 mL of the DPPH solution was added to 0.1 mL of dewaxed cerumen powder at a 20 mg/mL concentration. The sample solutions were incubated in the dark for 30 min, and the absorbance was measured in a UV-vis spectrophotometer (Agilent Cary 60, Santa Clara, CA, USA) using an advanced reads program at 517 nm. The samples were measured against a blank (2.9 mL of 96% ethanol added to 0.1 mL of DI water). Each sample was carried out in triplicate at room temperature. The control solution consisted of 2.9 mL of DPPH solution and 0.1 mL of DI water. Where A sample is the absorbance of the sample at 517 nm after 30 min, and A blank is the absorbance of the control. Following Equation (2), the total antioxidant (DPPH%) is calculated.
(2)$$\mathrm{DPPH}\;\left(\%\right)=\frac{A\;\mathrm{blank}-\;A\;\mathrm{sample}}{A\;\mathrm{blank}}\times 100$$

#### 2.7.2. Total Phenolic Content by Folin–Ciocalteu

The total phenolic content method was adapted from [18]. Stock solutions of the dewaxed cerumen powders in DI water at 20 mg/mL were prepared in 25-mL glass beakers. The solutions were vortexed (VELP Scientific ZX3 Advanced Vortex Mixer, Deer Park, NY, USA) at 1800 rpm for 1 min to break up the clumps. Then, 1 mL of stock solutions were diluted 10-fold in 96% ethanol and vortexed for 10 s. The diluted stock solutions of 1 mL were added to 1 mL 96% of ethanol and 5.5 mL of DI water in a 15-mL centrifuge tube. The blank was 1 mL of 96% ethanol instead of 1 mL of diluted solutions. Then, 1.25 mL of Folin–Ciocalteu reagent (diluted 1:2 in DI water) was added to the solutions, mixed by a vortex mixer for 30 s and incubated for 4 min. After this, 1.25 mL of 20% (*w*/*v*) Na_2_CO_3_ was added to stop the reaction. The final volume of each sample solution was 10 mL.

These solutions were left in the dark for 30 min and then centrifuged at 5000 RCF for 10 min (Thermo Scientific Sorvall ST 16, Waltham, MA, USA) to remove the salts formed after the addition of Na_2_CO_3_. The absorbance of the solutions was measured in a UV-vis spectrophotometer (Agilent Cary 60) using an advanced reading program at 765 nm against a blank. The average reading per sample is 0.2 s. The tests were conducted in triplicate at room temperature. The results were quantified based on a linear regression against the standard curve of y = 6.9062x + 0.1046, which had a correlation coefficient (R) of 0.962. The total phenolic contents were expressed as mg gallic acid equivalents (GAE)/g dry basis dewaxed cerumen.

### 2.8. Encapsulation Efficiency

The encapsulation efficiencies of the dewaxed cerumen powders were determined using the Folin–Ciocalteu method according to [21], applying Equation (3). TPC_Sample_ is the total phenols in the freeze-dried sample, which was derived by substituting the sample’s absorbance into the regression equation obtained from a standard curve of gallic acid. TPC_All_ is the total phenolic content of non-freeze-dried dewaxed cerumen after drying. All the measurements were carried out in triplicate.
(3)$${\mathrm{EE}}_{\mathrm{TPC}}=\frac{{\mathrm{TPC}}_{\mathrm{Sample}}}{{\mathrm{TPC}}_{\mathrm{All}}}\times 100$$

The yield of freeze-dried powders was calculated according to Equation (4) [18].
(4)$$\mathrm{Yield}\;\left(\%\right)=\frac{\mathrm{weight}\;\mathrm{of}\;\mathrm{encapsulated}\;\mathrm{dewaxed}\;\mathrm{cerumen}\;\mathrm{powder}\;\mathrm{after}\;\mathrm{freeze}\;\mathrm{drying}}{\mathrm{dry}\;\mathrm{weight}\;\mathrm{of}\;\mathrm{dewaxed}\;\mathrm{cerumen}\;\mathrm{and}\;\mathrm{carrier}\;\mathrm{in}\;\mathrm{encapsulated}\;\mathrm{solution}\;\mathrm{before}\;\mathrm{freeze}\;\mathrm{drying}}\;$$

### 2.9. Statistical Analysis

One-way analysis of variance (ANOVA) was performed using Minitab^®^ 18 Statistical Software at a confidence level of 95%. Tukey’s multiple comparison tests were performed and used for significant differences at the probability level of *p* = 0.05. The results were expressed as the mean +/− standard deviation.

## 3. Results

### 3.1. Properties of Dewaxed Cerumen Powder

The summary of the powders’ physical properties is shown in Table 1. The yield of dewaxed cerumen powder exceeded 80%. It is demonstrated that the freeze-drying process effectively provides a relatively high yield due to minimizing losses under vacuum and at low temperatures.

#### 3.1.1. Moisture Content

According to the results presented in Table 1, the mean concentration (MC) of the samples did not exhibit a statistically significant difference (*p* < 0.05). The moisture percentage of the samples were within the range of 1.5–2.3%. The MG:DA composite with a 3:7 ratio and a carrier concentration of 40% exhibited the least amount of moisture content, whereas the MG:DA composite with the same ratio but a carrier concentration of 20% had the highest moisture content.

#### 3.1.2. Water Activity and Glass Transition Temperature

The water activity of the freeze-dried powders ranged from 0.032 to 0.142 Aw, showing significant differences among the samples (*p* < 0.05). Among all the freeze-dried dewaxed cerumen powders, only 40% of the carrier concentration with maltodextrin: gum arabic (MD:GA) (9:1) has the lowest water activity of 0.032 among all the other samples. The increased concentration of carrier agents from 20 to 40% showed an overall decrease in water activity and a higher Tg. From Table 1, the water activities of 20% and 30% carrier concentrations of MD:GA (9:1) and 20% MD:GA (3:7) with the respective Aw of 0.108, 0.126 and 0.142 have no significant difference from the control of 31.25% MD at 0.113 Aw. These samples show that the presence of a higher MD concentration in the matrix led to higher water activity. With the increase in concentration from 30 to 40%, the Aw of ratio carriers MD:GA (9:1) and (3:7) reduced from 0.126 to 0.032 and 0.103 to 0.067, respectively.

The glass transition temperatures of encapsulated powders were found to be in the range of 16.7–26.2 °C as shown in Table 1. The concentration of carrier and varying ratios of MD and GA did not have any apparent influence on the Tg of samples, as the MD:GA (3:7) ratio did not increase Tg values when GA was higher in the ratio. Nonetheless, the glass transition temperature of the samples was higher than that of the unencapsulated dewaxed cerumen paste (−30 ± 3 °C) before freeze-drying. The Tg of samples has a linear relationship with their Aw, as shown in Figure 1.

#### 3.1.3. Freeze-Dried Powder Morphology

The morphology of all freeze-dried samples had flake-like structures and irregular crystalline-like shapes with sharp edges, as shown in Figure 2. At 20% carrier concentration across all ratios of carriers, microcapsules showed relatively smooth surfaces on the crystalline structure. In particular, the carrier ratio of 9:1 showed the smoothest surface and a crystalline structure with little porosity. At increasing carrier concentrations from 20 to 40% for respective MD:GA carrier ratios of 9:1, 5:5 and 3:7, the flake-like structure of particles was lessened with rougher surfaces and more porosity within the particles.

#### 3.1.4. Water Dispersibility (WD)

From Table 1, the WD of the samples was found to have no significant difference (*p* < 0.05). The WD of samples was in the range of 76.4–84.7%. The variability observed among the samples can be explained by the dispersibility of MD:GA within the samples. Nevertheless, the concentration of carriers did not influence the water dispersibility of the samples.

#### 3.1.5. Particle Size Distribution D (4,3)

The volume means diameter, D (4,3) of the samples’ particle size, ranges from 3.5 to 20.2 μm, showing significant differences among the samples (*p* < 0.05). Samples 31.25% MD, 20% MD:GA (9:1), 20% MD:GA (5:5) and 40% MD:GA (9:1) were found to have no significant difference with particle sizes > 12.2 μm. Powdered solutions contain two peaks, smaller particle sizes < 4 μm and larger particle sizes of 8–200 μm. All concentrations and ratios of carriers except 31.25% MD and 20% MD:GA (9:1) showed no significant difference among samples in the range of 3.5–13.2 μm.

#### 3.1.6. Microscope

In Figure 3, the dewaxed cerumen, when dissolved in water, had a self-assembled emulsion structure.

### 3.2. Antioxidant Activity of Dewaxed Cerumen Powders

#### 3.2.1. Antioxidant Activity

The summary of the powders’ antioxidant activity is given in Table 2. The inhibition of free radicals among samples ranged between 57.63–80.83%. The antioxidant activity for the carrier ratio MD:GA (3:7) was the lowest compared to MD:GA (9:1) and MD:GA (5:5) for all carrier concentrations of 20%, 30% and 40%. Despite this, the antioxidant activities among all the samples except the 31.25% MD showed no significant difference (*p* < 0.05).

#### 3.2.2. Total Phenolic Content (TPC) and Encapsulation Efficiency (EE)

TPC and EE% of samples had the same relationship, with values ranging from 3.1 to 5.1 mg GAE/g and 46.4 to 77.6 EE%, respectively. Before encapsulation, the total phenolic content of dewaxed cerumen was found to be 6.6 mg GAE/g which was higher than the freeze-dried encapsulated samples. The effects of carrier concentration and MD:GA ratios do not appear to interact. 31.3% MD was found to be significantly different from the other samples, with the highest TPC of 5.1 mg GAE/g and an encapsulation efficiency of 77.6%. The higher MD concentration in the carrier ratio, MD:GA (9:1), showed a decrease in TPC from 4.3 mg GAE/g for 20% carrier concentration to 3.8 mg GAE/g for 30% and 40% carrier concentration, and encapsulation efficiencies of 65.3%, 57.7%, and 57.2% with increasing concentrations from 20%, 30% and 40%, respectively. At a higher ratio of gum arabic, MD:GA (3:7), the phenolic content present was the lowest compared to MD:GA (9:1) and MD:GA (5:5) across all carrier concentrations. It showed a significant difference in concentration from 20–30% with the increase in TPC from 3.1–4.2 mg GAE/g, but the increase in concentration from 30–40% showed no significant difference with a TPC of 4.3 mg GAE/g. Conversely, MD:GA (5:5) showed relatively high TPC across increasing concentrations among the various ratios. MD:GA (5:5) showed no significant difference at 20% and 30% concentrations with 4.3 mg GAE/g and 4.1 mg GAE/g, respectively, but showed a significant difference at 40% with the increase in TPC of 4.5 mg GAE/g. At the second-highest EE% of 68.4%, the presence of maltodextrin and gum arabic in an even ratio provided the best protection of the antioxidant activity in the core material compared to 31.25% MD.

## 4. Discussion

### 4.1. Properties of Dewaxed Cerumen Powder

Fazaeli et al. (2012) [22] reported that powders produced with MD and GA combinations had yields greater than 80%. Among respective carrier concentrations, MD:GA (5:5) and MD:GA (3:7) yielded 90.6% and 99.7%, respectively, for 20% carrier concentration, compared to 31.25% MD yielding 88.1%. For 30% carrier concentration, no carrier ratios were higher than the control of 31.25% MD, where MD:GA (5:5) had the lowest yield of 83.5%, followed by MD:GA (3:7) of 86.5% and MD:GA (9:1) of 88.0%. For a 40% carrier concentration, only MD:GA (5:5) of 99.4% was higher than 31.25% MD of 88.1%. MD:GA (9:1) had the lowest yield of 81.9%, followed by MD:GA (3:7) of 83.1%.

#### 4.1.1. Moisture Content

Papoutsis et al. (2018) [21] also revealed that freeze-dried lemon by-products made with maltodextrin, soy protein, and i-carrageenan had a range of 1.2% to 2.2%. The powders’ MC was in the target moisture content range of 0.5–3% for fully freeze-dried products. The changes in MD and GA carrier ratios did not influence the MC of the samples. For MD:GA (9:1) and MD:GA (3:7), the increase in concentration caused a decrease in moisture content due to the reduced hygroscopicity of the powder [23]. For MD:GA (5:5), the moisture content increased from 1.9% to 2.2% due to the increased moisture adsorption and carriers’ molecular weight [24]. Ferrari et al. (2013) [25] reported a similar observation in the drying of blackberry powders, where the combination of 3.5% maltodextrin and 3.5% gum arabic produced a higher moisture content than 7% maltodextrin.

#### 4.1.2. Water Activity and Glass Transition Temperature

Water activity is an important parameter to consider, as it determines the shelf life of powdered products. When lower than 0.6 Aw, the food powders show that they are all microbiologically and chemically stable [23]. The water activity of freeze-dried date powder went in the same direction as the concentration of the carrier [23]. Our results were comparable to those that [26] observed. The increase in Aw was reported with the rise in MD concentration from 0% to 20%, which could be due to the different water-binding capacities of the carriers caused by their chemical structure as polyols [23]. In each hydroxyl group of maltodextrin, hydrogen bonding may occur and bind to more than one water molecule [26]. Hence, samples with higher maltodextrin have higher water retention, leading to higher water activity [22]. Though 20% MD:GA (3:7) had a lesser presence of MD and was not significantly different from 31.25% MD, it had a high Aw and Tg of 0.142 and 22 °C, respectively, compared to 31.25% MD’s Tg of 16.7 °C. The high ratio of gum arabic in the matrix did not reduce the matrix’s water activity but increased the Tg of the powdered samples. The reduced Aw with increased concentration could be due to the increased encapsulation effect, which diminishes the ability of the core material to bind onto more water molecules [27]. The noticeably low Aw of MD:GA (9:1) could be attributed to maltodextrin’s high total solid content [28]. MD:GA (5:5) and had no significant difference in Aw among the samples with increasing concentration due to multiple factors such as the conformation and topology of the molecule and the hydrophilic/hydrophobic sites absorbed at the interface [29]. The combination of MD:GA at an even ratio with increasing concentration could have increased the solid’s content and its emulsifying properties.

Tg is the temperature at which an amorphous system goes from being glassy to rubbery. Above this temperature, the physical and chemical properties of the powdered samples will change, causing them to stick, cake, collapse, lose volatiles, and oxidize [30]. In order to prevent the molecular mobility of the matrix from accelerating, it is possible to extend the stability of powders by storing the samples at a lower temperature than their Tg [31]. This result does not agree with Seerangurayar (2017) [23], who reported that variations with the increased presence of GA (66 ± 6 °C) with high Tg compared to MD (55 ± 2 °C) produced powders with higher Tg values. The increase in water activity and the reduction of glass transition temperature because of the critical water activity change with the glassy structure’s molecular weight [30]. The plasticizing effect of water decreases the Tg of samples [32].

#### 4.1.3. Freeze-Dried Powder Morphology

Observations by Papoutsis et al. (2018) [21] and Eratte et al. (2015) [33] were comparable to the morphology of our freeze-dried samples. The asymmetrical shape has resulted from the homogenization of the core material in the matrix solution [34] and the low temperatures of the freeze-drying. The lack of forces to break up the ice crystals into droplets substantially changed the surface topology during the lyophilization process [26,35]. The presence of higher carrier concentrations in propolis led to an increase in core materials disseminated in the wall matrix, resulting in rougher surfaces. Consequently, the flake-like structure of particles decreased while porosity inside the particles increased [33]. The morphology of increased GA is comparable with the Kaushik and Roos (2007) examination, where the freeze-dried matrix is free of dents and shrinkages. These amorphous glasses also help protect against light and oxygen [36]; they are more hygroscopic, and the ability to release the core material results in a more readily dissolved solution [33]. The increase in porosity is due to the sublimation of smaller ice crystals during freeze-drying [37]. These pores also facilitate the rehydration of the product, allowing for rapid and effective dissolution [38]. Hence, this supports the high water dispersibility of the freeze-dried samples.

#### 4.1.4. Water Dispersibility (WD)

The water dispersibility for MD:GA (9:1) was expected to rise with increasing carrier concentration due to the higher presence of MD solubilized in the solution [18]. As dewaxed cerumen is naturally insoluble in water because of its hydrophilic and hydrophobic compounds, the use of MD and GA successfully increased the dispersibility of dewaxed cerumen in powder. Pratami et al. (2020) [39] have also reported similar results, where propolis encapsulated with MD and GA was in the range of 73.7–91.5%.

#### 4.1.5. Particle Size Distribution D (4,3)

Through physical or electrostatic interaction during the homogenization process, wax and resin-composed material with a small fraction of essential oils present can be hypothesized to have formed emulsions and the resin-composed material to have formed microcapsules. As the MD and GA dissolve readily in water, the naturally hydrophobic propolis fraction is exposed to water and has been seen to sediment into the solution over time. These particles may be responsible for the second peak identified in the particle size distribution due to the formation of coalescence or flocculation [40]. The effect of concentration and ratios of carriers did not have a noticeable influence on the particle size distribution. However, the presence of gum arabic in the powders achieved a smaller D (4,3) particle size caused by the protective film of gum arabic, which increases the viscosity of the continuous phase and prevents the aggregation between larger particles [40]. The carrier concentrations of 30% and 40% displayed smaller particle sizes compared to 20%, possibly due to the increased coating material encapsulating the propolis, which prevents the combination of particles [41]. Therefore, smaller particle sizes were obtained.

#### 4.1.6. Microscope

When the dewaxed cerumen was dissolved in water, an emulsion structure self-assembled. The chemicals with polar and non-polar properties present in the propolis matrix effectively capture and enclose bubbles within the particles, resulting in the formation of a bilayer membrane composed of phospholipids [42]. These different substances change the polarity of propolis, rendering it insoluble in water and increasing its affinity for incorporation into its phospholipidic bilayer membrane [42]. 

### 4.2. Antioxidant Activity of Dewaxed Cerumen Powders

#### 4.2.1. Antioxidant Activity

The antioxidant activity of the carrier ratio MD:GA (3:7) was the lowest compared to the carrier ratios MD:GA (9:1) and MD:GA (5:5) for all carrier concentrations of 20%, 30%, and 40%. This could be because the freeze-drying process was done at very low temperatures, which could have caused the gum arabic proteins to denature in the cold [43]. Since the freeze-drying process could not be controlled, the sample matrices may have been exposed to dehydration stresses that made it more difficult for them to hold the high antioxidant compounds of propolis. The results show that the presence of gum arabic as a wall material has prevented the loss of volatiles in contact with the atmosphere [44]. *Hibiscus acetosella* extract from freeze-dried powders constituted with gum arabic provided good bioactive component protection and retention, according to [45]. In the freeze-dried dewaxed cerumen powders, gum arabic contributed to the emulsifying and film-forming properties of the carrier, which better encapsulated antioxidant compounds.

#### 4.2.2. Total Phenolic Content (TPC) and Encapsulation Efficiency (EE)

The amount of TPC sustained in the freeze-dried powders was reduced. The cause of this is due to the drying process of the ethanolic solvent, which subjected the ethanolic dewaxed cerumen to heat and light. Other possible causes are the scattering of substances in the process of microcapsule formation [46], dehydration stresses during the freeze-drying process, exposure to light and oxygen during and post-lyophilization [21], and the leakage of sensitive compounds from the dewaxed cerumen matrix into the sublimated ice crystals during the freeze-drying process [45]. Hence, the reduced amount of total phenolic content after the freeze-drying process was anticipated. Our result is comparable with González-Ortega et al. (2020) [35], where maltodextrin as the only wall component resulted in a higher retention of phenolic compounds. The combinations of MD:GA were lower than MD alone, with an EE% range of 46.4–68.4%, as shown in Table 2. The increased solid content of wall materials relative to core materials could lead to adverse effects on bioactive components due to the higher solid content and lower water content of the matrix. The water’s ability to evaporate decreases during the freeze-drying stage, which could have placed more stress on the bioactive components available, resulting in less protection of the core material [40]. The results did not agree with [47], where the combination of MD:GA (100:1) showed a higher phenolic content maintained than pure maltodextrin. The small percentage of gum arabic present as a coating material was to protect the phenolic content with its highly branched structure [47]. However, the polyphenol complex with polysaccharides could be affected by the solubility, molecular size, shape and mobility of polyphenols [48]. Thus, explaining the variation of polyphenol entrapment. Our results were similar to Sukri et al. (2020) [47], where the increased concentration of gum arabic to maltodextrin higher than MD:GA (100:1) reduced the amount of phenolic content maintained. Due to the possible cold denaturation of proteins in gum arabic at −80 °C storage and freeze-drying, the structural changes may have caused the interactions between the polyphenolic substances and polysaccharides to diminish [43]. Thus, it provided less protection for bioactive components. The bulking properties of maltodextrin protected the proteins available in gum arabic from cold denaturation. Hence, allowing the plasticity of gum arabic to form a good film over the core material and giving good encapsulation efficiency [49].

## 5. Conclusions

The microencapsulation of dewaxed cerumen using MD and GA through freeze-drying provided high yields and good physical properties of the powders. The powders’ antioxidant activity was able to be retained, but the cold denaturation of proteins during the freeze-drying process can weaken the protection of dewaxed cerumen. The combination of carriers and concentration significantly affects water activity, particle size in solution, particle morphology, antioxidant activity, and encapsulation efficiency. The carrier concentration of 40% achieved low water activities, the highest glass transition temperatures, the highest antioxidant activities and good encapsulation efficiency. Likewise, the carrier ratio of MD:GA (5:5) across the various concentrations has achieved powders with high yields, good physical properties, and high antioxidant activity.

The use of cerumen from stingless bees in the food industry remains relatively new and little known. However, there is potential for further research and implementation of stingless bee cerumen in food products. It is necessary to investigate its functional properties and active ingredients to maximize the benefits of stingless bee cerumen [5,50]. Moreover, the development of suitable processing techniques for extracting, purifying, and incorporating cerumen into various food items opens new possibilities. Additionally, it is important to ensure that any use of cerumen in the food industry complies with food safety regulations and considers the preservation of this pollinator.

## Figures and Tables

**Figure 1 foods-12-03740-f001:**
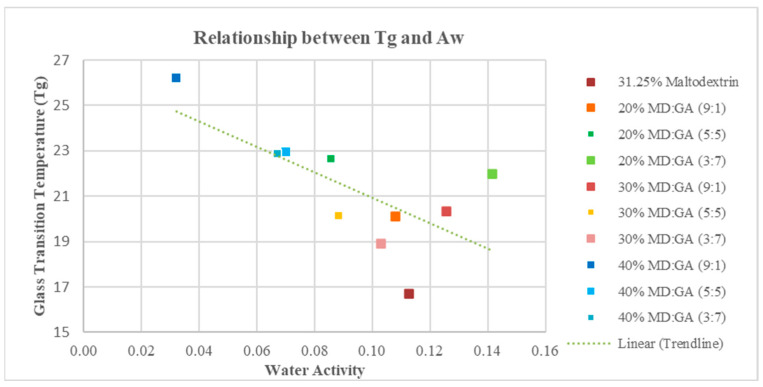
Glass transition temperature of freeze-dried dewaxed cerumen at respective water activities.

**Figure 2 foods-12-03740-f002:**
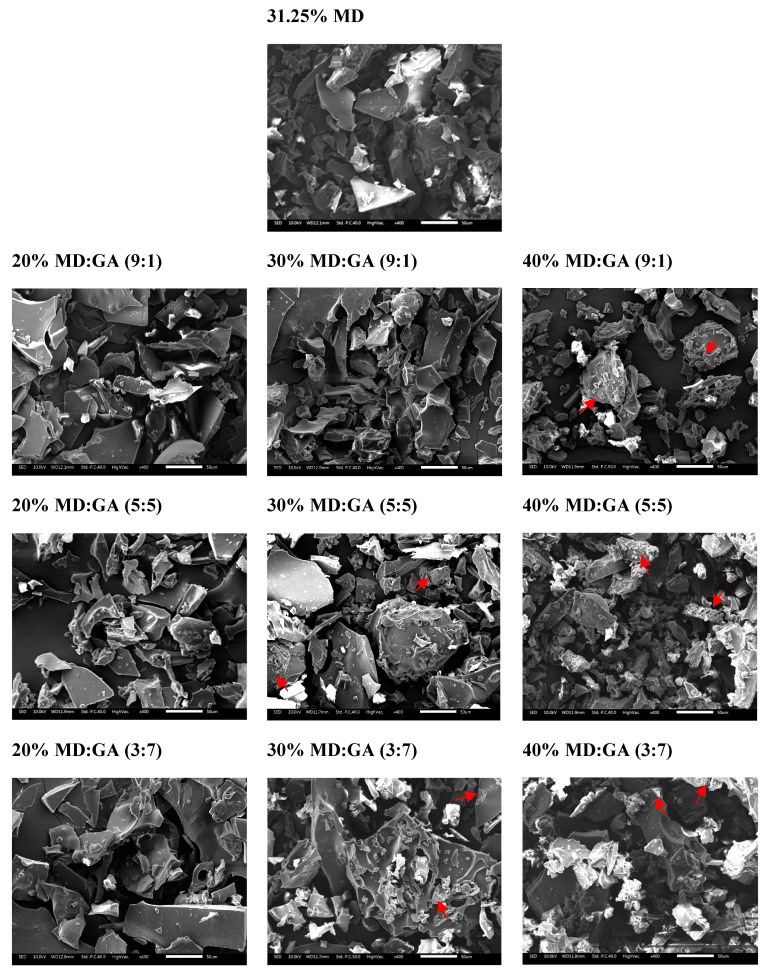
Morphology of freeze-dried dewaxed cerumen (400× magnification) at different MD:GA ratio. Red arrows indicate the porosity in the particles.

**Figure 3 foods-12-03740-f003:**
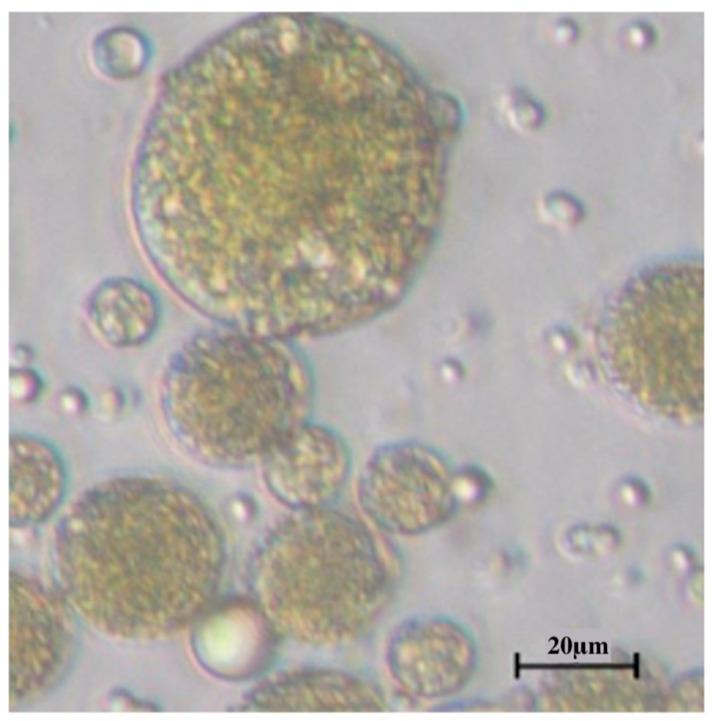
Example of dewaxed cerumen particles (100×).

**Table 1 foods-12-03740-t001:** A summary result of physical properties of dewaxed cerumen powder.

Sample	Carrier Ratio (MD: GA)	Parameters (Mean ± SE)					
Carrier Conc. (%)		Yield (%)	MC (%)	WA (a_w_)	GTT (°C)	WD (%)	PS D (4,3) (μm)
31.25	10:0	88.09	1.73 ± 0.01 ^a^	0.1127 ± 0.0069 ^abc^	16.70 ± 0.99 ^a^	80.92 ± 10.01 ^a^	14.20 ± 2.97 ^ab^
20	9:1	84.63	1.93 ± 0.00 ^a^	0.1081 ± 0.0093 ^abc^	20.10 ± 0.57 ^a^	76.40 ± 0.21 ^a^	20.20 ± 5.66 ^a^
	5:5	90.64	1.89 ± 0.01 ^a^	0.0857 ± 0.0130 ^cde^	22.65 ± 1.63 ^a^	77.67 ± 7.32 ^a^	13.20 ± 2.55 ^abc^
	3:7	99.72	2.28 ± 0.65 ^a^	0.1417 ± 0.0156 ^a^	22.00 ± 3.25 ^a^	84.72 ± 2.02 ^a^	4.53 ± 0.02 ^bc^
30	9:1	87.96	2.08 ± 0.40 ^a^	0.1257 ± 0.0052 ^ab^	20.35 ± 1.20 ^a^	80.75 ± 2.97 ^a^	4.31 ± 0.07 ^bc^
	5:5	83.49	2.10 ± 0.17 ^a^	0.0884 ± 0.0081 ^cde^	20.15 ± 1.63 ^a^	77.35 ± 0.64 ^a^	8.92 ± 2.09 ^bc^
	3:7	86.47	2.05 ± 0.23 ^a^	0.1029 ± 0.0033 ^bcd^	18.90 ± 0.14 ^a^	77.82 ± 6.75 ^a^	9.37 ± 0.01 ^bc^
40	9:1	81.92	1.50 ± 0.21 ^a^	0.0321 ± 0.0058	26.20 ± 1.41 ^a^	83.03 ± 2.02 ^a^	12.21 + 4.09 ^abc^
	5:5	99.38	2.17 ± 0.04 ^a^	0.0699 ± 0.0082 ^de^	22.95 ± 5.73 ^a^	82.90 ± 0.35 ^a^	3.80 ± 0.47 ^bc^
	3:7	83.07	1.45 ± 0.08 ^a^	0.0671 ± 0.0001 ^ef^	22.90 ± 4.67 ^a^	83.33 ± 9.37 ^a^	3.49 ± 0.64 ^c^

Means followed by different superscripts within column are significantly different (*p* < 0.05); Carrier con. = Carrier Concentration, MC = Moisture Content, WA = Water Activity, GTT = Glass Transition Temperature, WD = Water Dispersibility, PS = Particle Size.

**Table 2 foods-12-03740-t002:** A summary of the result of antioxidant activity and encapsulation efficiency of dewaxed cerumen powder.

Sample		Antioxidant Activity (%Inhibition)	Total Phenolic Content (TPC) (mg GAE/g)	Encapsulation Efficiency (EE%)
Carrier Conc. (%)	Carrier Ratio (MD:GA)			
31.25	10:0	57.63 ± 5.57 ^b^	77.63 ± 0.28 ^a^	5.14 ± 0.01 ^a^
20	9:1	79.97 ± 0.73 ^a^	65.34 ± 0.19 ^c^	4.33 ± 0.02 ^c^
	5:5	74.71 ± 1.42 ^a^	64.40 ± 2.90 ^cd^	4.26 ± 0.20 ^cd^
	3:7	74.05 ± 9.71 ^ab^	46.43 ± 0.08 ^f^	3.07 ± 0.01 ^f^
30	9:1	71.81 ± 5.45 ^ab^	57.70 ± 0.07 ^e^	3.82 ± 0.02 ^e^
	5:5	77.13 ± 0.65 ^a^	61.86 ± 0.36 ^d^	4.09 ± 0.03 ^d^
	3:7	69.26 ± 6.27 ^ab^	63.69 ± 0.06 ^cd^	4.21 ± 0.01 ^cd^
40	9:1	80.83 ± 3.93 ^a^	57.17 ± 0.20 ^e^	3.78 ± 0.02 ^e^
	5:5	80.51 ± 4.40 ^a^	68.41 ± 0.25 ^b^	4.53 ± 0.03 ^b^
	3:7	70.73 ± 9.81 ^ab^	65.07 ± 0.02 ^c^	4.31 ± 0.01 ^c^

Means followed by different superscripts within column are significantly different (*p* < 0.05). Carrier con. = Carrier Concentration.

## Data Availability

The data used to support the findings of this study can be made available by the corresponding author upon request.

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
