# Peer review of "Effects of Maltodextrin and Gum Arabic Composition on the Physical and Antioxidant Activities of Dewaxed Stingless Bee Cerumen"

_foods, 2023, doi:10.3390/foods12203740_

Round 1

Reviewer 1 Report

Find below my comments to improve the manuscript.

Provide the list of abbreviations.

Line 130

Provide the following rotary evaporator conditions

vacuum pressure, heating bath temperature, vapor temperature, coolant temperature, and rotating speed.

Line 137

Why select these ratios?

Line 179

Why use duplicates for analytical measurements rather than triplicates?

Line 180

Why didn’t you measure the zeta potential too?

Line 201

Why only dpph assay?

The ABTS and DPPH assay is an excellent tool for evaluating hydrophilic and hydrophobic antioxidants, the CUPRAC assay for determining the suppression of OH radical synthesis, and the potential of an antioxidant to reduce ferric (Fe3+) to ferrous (Fe2+) ion by transferring electrons could be assessed using the RP assay., etc.

In the SEM image, use red arrows to show in the image what you are explaining, this will help the readers to better understand.

The discussion needs further improvements as well. Especially compare the results in this study with most recent related literature and do a critical evaluation on the trends/similarities/differences

Line 481

You can run a principal component analysis if you have the ability. If you don’t, you can leave that.

PCA reduces the matrix dimensionality and identifies specific correlation patterns while retaining most information.

The manuscript article requires revision in grammar, sentence structure, and reference format. Overused Passive voice in the manuscript seems hard to read. Please carefully check the sections: introduction, results, discussion and conclusions. Please try to reword the phrases in the active voice. Grammar and punctuation mistakes. For consistency, please use the manuscript in just one English style (a non-variant British or British style, American style, etc.). There are phrases with the verb in the wrong tense. Sentences with words misspelt. Words overused or unnecessary. Nouns without determiner or unnecessary.

Minor revision

Author Response

I would like to express my gratitude for your time and dedication in reviewing the manuscript. You can find my response in the attached PDF file, with the relevant changes highlighted in red within the text (word file).

Reviewer 2 Report

The authors have studied the use of maltodextrin and Arabic gum to prepare particles with the cerumen of stingless bees, particularly focusing on its polyphenolic content and physical properties. While the manuscript holds potential for the food industry, it has several issues that need rectification:

  1. Abstract: Punctuation in the abstract requires correction. Furthermore, the objective of the study should be clearly outlined.
  2. Introduction: The first paragraph provides extensive information on cerumen and bees, yet abruptly transitions to propolis in the second paragraph. The authors should clarify this sudden switch. Additional references regarding cerumen extraction, its biological properties, and applications should also be incorporated.
  3. Objective Clarification: The manuscript should elucidate the primary goals of the study, answering specific questions like:
    • Why was there a need to encapsulate cerumen?
    • What are the reasons for not using dewaxed material in food products?
    • Why were maltodextrin and Arabic gum chosen as the carriers?
  4. Methods, Section 2.3: The section refers to propolis. It is crucial to clarify if propolis was utilized in this study.
  5. Results: The table should be repositioned closer to its initial citation in the text, ensuring better flow and coherence for readers.
  6. Figure 1: The figure should display the linear equation, and the relationship between water activity and the glass transition temperature should be emphasized to allow assessment of linearity.
  7. Section 4-1.3: The discussion pivots back to propolis. The matrix used in the referenced studies should be specified in the text to ensure clarity and coherence.
  8. Implications: The manuscript should conclude by emphasizing the final implications of the methods employed. This should encompass:
    • Their potential utility for the food industry.
    • Practical applications from the research findings.

The overall language usage in the manuscript is adequate.

Author Response

(The authors gave the same response as above.)

Reviewer 3 Report

This is an interesting research. However, it requires some revisions.

1. The authors should add city and country names after company. For example, line 95, Ingredion; line 98, Sigma-Aldrich, and so on.

2. Page 3, line 101: It is suggested the authors can add reference number [20] after Krell (1996). Also the same issue is in line 121, Kubiliene et al. (2015).

3. Page 5, line 209: A mild suggestion, sample and blank of Asample and Ablank can be subscripted like equation 3 (page 6, line 237).

4. Page 6, equation 4: please move ‘Yield (%)’ to the left.

5. Page 6, Result 3.1.1 moisture content: this paragraph is too short.

6. Page 7, Figure 2: Is it possible to bring the legend from page 8 to page 7, so that the figure and length can stay together.

7. Page 8, Result 3.1.6. microscope: this paragraph is too short.

8. Page 8, Figure 3: To me, the number/word of the scale is invisible. Please enlarge it.

9. Page 13, line 502: The first two sentences of Author Contributions should be removed.

10. Page 13, line 507: Please provide funding numbers.

Author Response

(The authors gave the same response as above.)

Reviewer 4 Report

Dear authors,

Please check once again the technical and grammatical errors…please express values with numbers in the text. Check the units once again in the whole manuscript!

No comments.

Author Response

(The authors gave the same response as above.)

Round 2

Reviewer 1 Report

accept

minor

Reviewer 3 Report

This manuscript is imporved.